# An Analysis of Abstracted Model-Based Reinforcement Learning

## Abstract

Many methods for Model-based Reinforcement Learning (MBRL) provide guarantees for the accuracy of the Markov decision process (MDP) model they can deliver. At the same time, state abstraction techniques allow for a reduction of the size of an MDP while maintaining a bounded loss with respect to the original problem. It may come as a surprise, therefore, that no such guarantees are available when combining both techniques, i.e., where MBRL merely observes abstract states. Our theoretical analysis shows that abstraction can introduce a dependence between samples collected online (i.e., in the real world), which invalidates most results for MBRLs in this setting. Collecting samples using a simulator can avoid this problem. We conclude that we should be careful when applying MBRL methods to abstracted real-world data.

## 1 Introduction

When trying to find good solutions to MDPs using Reinforcement Learning (RL) a fundamental problem is the exploration-exploitation dilemma: when to take actions to obtain more information, and when to take actions that maximize reward based on the current knowledge. Tabular MBRL methods have found good ways to deal with this dilemma [7, 28, 14].

However, MDPs can be very large, which can be problematic for these methods. One way to deal with this is to reduce the size of the MDP. State abstractions are one way to do this [17, 1]. We are interested in approximate state abstractions since they allow for greater reductions of the MDP, though there is a trade-off with solution quality [1]. Specifically, we assume we have an *approximate model similarity abstraction function* $\phi$ [1] that maps states to abstract states. The environment returns states $s$, but the agent receives $\phi(s)$, see Figure 1. This setting, which was considered before [22, 2], is what we call *Abstracted RL*, and is the topic of this paper.

Abstracted RL corresponds to RL in a Partially Observable MDP (POMDP), as previously described [5]. It is well known that policies for POMDPs that only base their action on the last observation $\phi(s)$ could be arbitrarily bad [26]. However, when we assume that $\phi$ is an approximate model similarity abstraction [1] this worst case may not apply: Based on the observed abstract states the agent learns an (empirical) abstract model. If we could show that this learned model

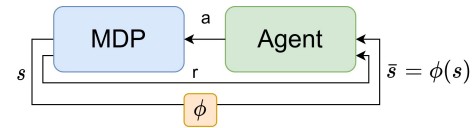

Figure 1: Abstracted RL, the agent observes $\bar{s} = \phi(s)$ instead of $s$. Image based on Abel et al. [2]

is close to an 'abstract MDP' (details in Section 2.2), we could give finite-sample guarantees on the performance in the original MDP by combining results from MBRL and abstraction.

However, in MBRL, to guarantee (with high probability) that the learned model is close to the actual environment model, it is typical (e.g., [28, 14]) to use concentration inequalities such as Theorem 2 from Weissman et al. [30]. But this theorem relies on independent and identically distributed (i.i.d.) samples for each state-action pair. In this paper, we analyze online collection of such samples in Abstracted RL and show that they are not independent[1], which means that *most guarantees for existing MBRL methods do not hold in the online Abstracted RL setting.*[2]

On the positive side, when we have access to a simulator, we show how this can be used to collect the data such that the typical MBRL guarantees hold and we can learn an accurate model. We discuss that emulating this in the real world is possible, but extremely sample inefficient, thus highlighting the difficulty of assuming that we would have access to an i.i.d. dataset, as in some earlier works.

## 2 Preliminaries

We assume the environment the agent is acting in can be represented by an infinite horizon MDP $M \coloneqq \langle S, A, T, R, \gamma \rangle$. Where $S$ is a finite set of states $s \in S$, $A$ a finite set of actions $a \in A$, $T$ a transition function $T(s'|s,a) = \Pr(s'|s,a)$, $R$ a reward function $R(s,a)$ which gives the reward received when the agent executes action $a$ in state $s$, and $\gamma$ the discount factor ($0 \leq \gamma < 1$).

In RL the goal of the agent is to find an optimal policy $\pi^* : S \rightarrow A$ which maximizes the expectation of the discounted cumulative reward. $V^\pi(s)$ denotes the expected value of the discounted cumulative reward under policy $\pi$ starting from state $s$. Similarly, $Q^\pi(s,a)$ denotes the expected value of the discounted cumulative reward when first taking action $a$ from state $s$ and then following policy $\pi$.

### 2.1 Model-Based RL

MBRL methods learn a model from the experiences, these are obtained by the agent acting in the MDP. The learned model is usually the empirical model, directly based on the experience the agent obtains [7, 28, 14]. Per state-action pair the agent stores the next-states reached after taking action $a$ from state $s$ in sequence $Y_{s,a}$: $Y_{s,a} : \{s'^{(1)}, s'^{(2)}, \cdots, s'^{(m)}\}$. We use $Y$ to refer to the collection of all $Y_{s,a}$. From this the empirical, or learned, model $T_Y$ is constructed, that just counts how often we have seen the transition to a next-state, and normalizes this:

$$\forall_{s' \in S} \ T_Y(s'|s,a) \triangleq \frac{1}{m} \sum_{i=1}^{m} \mathbb{1}\{Y_{s,a}^{(i)} = s'\}, \tag{1}$$

where $\mathbb{1}\{\cdot\}$ denotes the indicator function of the specified event, i.e., it is 1 if $Y_{s,a}^{(i)} = s'$ and 0 otherwise.

To give finite-sample guarantees on the accuracy of the estimate $T_Y$,[3] concentration bounds such as Theorem 2.1 from Weissman et al. [30] are often used, e.g. in Strehl and Littman [28], Jaksch et al. [14]. However, these typically make use of the fact that samples are i.i.d. In most MBRL settings this is not a problem under some assumptions, e.g. when the MDP is communicating [25]. In this case due to the Markov property the obtained samples are i.i.d.

In general, of course the hope is that with enough samples the learned model $T_Y$ becomes accurate. With accurate we mean that the distance between $T_Y(\cdot|s,a)$ and $T(\cdot|s,a)$ will be small, where the distance is measured using the $L_1$ norm, defined as:

$$||T_Y(\cdot|s,a) - T(\cdot|s,a)||_1 \triangleq \sum_{s' \in S} |T_Y(s'|s,a) - T(s'|s,a)|. \tag{2}$$

Part of theorem 2.1 from Weissman et al. [30], slightly reworded, then gives a guarantee of accuracy:

---

[1]We also show that samples are not identically distributed, but demonstrate that that problem would be resolvable.

[2]The reader might be puzzled by this statement, since certain guarantees on the combination of abstraction and RL are known. This can be explained by the generality of Abstracted RL: in this setting there is a non-stationarity caused by the clustering of states with different dynamics. There is a lot of related work in other abstraction settings (e.g., state aggregation) where this complication does not occur due to the particularities of their setting [24, 11, 19, 20, 23, 10]. In section 4 we give details to back up our claim for individual papers.

[3]This is a crucial element in being able to guarantee good performance, where performance can be measured in different ways, e.g. in PAC-MDP terms [28] or in terms of regret [14]. We focus on the model quality.

**Lemma 1** ($L_1$ inequality [30]). *Let $\mathbf{Y}_{s,a} = Y^{(1)}, Y^{(2)}, \cdots, Y^{(m)}$ be i.i.d. random variables distributed according to $T(\cdot|s,a)$. Then, for all $\epsilon > 0$,*

$$\Pr(||T_Y(\cdot|s,a) - T(\cdot|s,a)||_1 \geq \epsilon) \leq (2^{|S|} - 2)e^{-\frac{1}{2}m\epsilon^2}. \tag{3}$$

In this way, MBRL can upper bound the probability that the learned model, based on $m$ samples, for a state-action pair $(s,a)$ $T_Y(\cdot|s,a)$ will be far away ($\geq \epsilon$) from the true model $T(\cdot|s,a)$.

The situation is more subtle if the MDP is not communicating, i.e., if there exists $s_1, s_2 \in S$ for which there is no deterministic policy that eventually leads from $s_1$ to $s_2$. This can create a dependence between the samples [28]. Intuitively this happens because, if we look at one particular state-action pair $(s,a)$, there might be a transition to state $s'$ such that the probability to return to $s$ is 0. Thus if we would have $n$ outcomes of $(s,a)$ we would immediately know that at least $n-1$ outcomes were not state $s'$. Since as soon as we observe $s'$, we know the agent would not be able to return to state $s$. Strehl and Littman [28] show that in this setting it is still possible to use Lemma 3 as an upper bound.

## 2.2 State abstraction for given models

In the planning setting, where the model is known a priori, a state abstraction can be formulated as a grouping or mapping from ground states to abstract states [18]. This is done with an abstraction function $\phi$, a surjective function that maps from ground states, $s \in S$, to abstract states $\bar{s} \in \bar{S}$: $\phi(s) : S \to \bar{S}$. Here $\bar{S}$ is defined as $\bar{S} = \{\phi(s)|s \in S\}$.We use the $^-$ notation to refer to the abstract space. We slightly overload notation and let $\bar{s}$ both denote an abstract state as well as the set of ground states that map to the abstract state $\bar{s}$, i.e., $\bar{s} = \{g \in S \mid \phi(g) = \bar{s}\}$, if $\bar{s} \in \bar{S}$. The use should be clear from the context. We define the probability to transition to an abstract state $\Pr(\bar{s}'|s,a)$ as follows:

$$\Pr(\bar{s}'|s,a) \triangleq \sum_{s' \in \bar{s}'} T(s'|s,a). \tag{4}$$

This is a very general form of state abstraction, that clusters together states with different dynamics into abstract states. Note that we do assume that the given state abstraction deterministically maps states to an abstract state. This in contrast to some related work on problems with block structure [10], where a Markov state can lead to multiple observations (abstract states in our terminology) that need to be aggregated appropriately to result in a small MDP [4, 10].

**Approximate model similarity abstraction**  Many different abstraction criteria exist [17], here we focus on approximate model similarity abstraction [1]. In this abstraction two states can map to the same abstract state if their behavior is similar, i.e., when the reward function and the transitions to abstract states are close. Approximate model-similarity is defined as follows:

**Definition 1.** *An approximate model-similarity abstraction, $\phi_{model,\eta}$, for fixed $\eta$, satisfies:*

$$\phi_{model,\eta}(s_1) = \phi_{model,\eta}(s_2) \implies \forall_a |R(s_1,a) - R(s_2,a)| \leq \eta,$$
$$\forall_{\bar{s}' \in \bar{S}, a} |\Pr(\bar{s}'|s_1,a) - \Pr(\bar{s}'|s_2,a)| \leq \eta. \tag{5}$$

From now on we will just refer to $\phi_{\text{model},\eta}$ as $\phi$.

We note that the abstraction we consider, approximate model-similarity abstraction, is still quite generic. It can cluster together states that have different transition and reward functions. However, in the online Abstracted RL setting, the differences in dynamics can cause a dependence between the samples, as we will show in detail in section 3. E.g. looking at $(\bar{s},a)$, the probability that we reach a state $s'$ depends both on the probability that we reach a particular state $s \in \bar{s}$ and then state $s'$ from $s$.

Returning to abstraction of a given model, it is possible to construct an abstract MDP $\bar{M}_\omega$ from the model of an MDP $M$ and an abstraction function $\phi$, where $\omega$ is an action-specific[4] weighting function, defined as follows:

---

[4]The action-specific weighting function is a more general weighting function than is typically used, e.g. by Li et al. [18], which is not action-specific, i.e., it only depends on the state $s$. More formally it is the case where $\forall_{a,a' \in A, \ s \in S} \ \omega(s,a) = \omega(s,a')$.

**Definition 2.** *We refer to the weight associated with a ground state, $s \in S$, and action, $a \in A$, by $\omega(s, a)$. We have:* $\forall_{s \in S, a \in A} \, 0 \leq \omega(s, a) \leq 1$ *and* $\sum_{s' \in \phi(s)} \omega(s', a) = 1$.

The weighting function can be used to create abstract transition and reward functions, which are a weighted average of the ground function. In this way, from $M$, $\phi$ and any $\omega$ we can *construct* an abstract MDP $\bar{M}_\omega$:

**Definition 3.** *Given an MDP $M$, $\phi$, and $\omega$, $\bar{M}_\omega = \langle \bar{S}, A, \bar{T}_\omega, \bar{R}_\omega, \gamma \rangle$ is constructed as:*

$$\bar{S} = \{\phi(s) \mid s \in S\}, A = A, \gamma = \gamma, \tag{6}$$

$$\forall_{\bar{s} \in \bar{S}, \, a \in A} \; \bar{R}_\omega(\bar{s}, a) = \sum_{s \in \bar{s}} \omega(s, a) R(s, a), \tag{7}$$

$$\forall_{\bar{s}, \bar{s}' \in \bar{S}, \, a \in A} \; \bar{T}_\omega(\bar{s}' | \bar{s}, a) = \sum_{s \in \bar{s}} \sum_{s' \in \bar{s}'} \omega(s, a) T(s' | s, a). \tag{8}$$

An abstract MDP $\bar{M}_\omega$ is just an $MDP$. This means we can use any planning method we like to find an optimal policy $\bar{\pi}^*$ for $\bar{M}_\omega$.

What we are interested in is the performance of a policy on the abstract space, when applied on the original problem $M$. Any policy on the abstract space $\bar{\pi}$ can be used in $M$ as follows $\bar{\pi}(s) \coloneqq \bar{\pi}(\phi(s))$, leading to $V^{\bar{\pi}^*}$. It has been shown that we can upper bound the loss in performance due to using an optimal policy for $\bar{M}_\omega$, $\bar{\pi}^*$ in $M$ instead of using the optimal solution for $M$ [8, 1, 29]:

**Lemma 2** (Lemma 4 [29])**.** *An approximate model similarity abstraction (Definition 1), has sub-optimality bounded in $\eta$:* $\forall_{s \in S} \, V^*(s) - V^{\bar{\pi}^*}(s) \leq \frac{2\eta + 2\gamma(|\bar{S}| - 1)\eta}{(1-\gamma)^2}$.

# 3  Abstracted MBRL and the problem of online data collection

As explained, we are interested in Abstracted RL, where we have an approximate model similarity abstraction function $\phi$ and an MDP $M$. The agent acts in $M$ but only observes $\phi(s)$ using abstraction function $\phi$, as in Figure 1. This setting can also be seen as a POMDP, where the states are hidden and there is a deterministic observation function, $o = \phi(s)$. However, in contrast to the usual POMDP settings, we look for a myopic (memoryless) policy. While we know that in general this can lead to arbitrarily bad results [26], in this case the value loss would be bounded in the *planning* setting by Lemma 2. However, now we assume we are in the Abstracted RL setting, and the result for planning may not hold for the learned model.

We assume we know $S, A, R, \gamma$, and $\phi$ (and thus $\bar{S}$), but do not know the transition function.[5] Since we do not know the transition function we can neither simply do planning on $M$ nor can we construct an abstract MDP, using Definition 3, and solve that. Instead, we let the agent interact with $M$ but use $\phi$ to let the agent observe $\phi(s)$, instead of $s$, and build a learned (abstract) model from the observations it obtains. We show the general Abstracted MBRL procedure in Algorithm 1.

The agent collects data for every abstract state-action pair $(\bar{s}, a)$, which is stored as sequences $\bar{Y}_{\bar{s}, a}$:

$$\bar{Y}_{\bar{s}, a} : \{\bar{s}'^{(1)}, \bar{s}'^{(2)}, \cdots, \bar{s}'^{(m)}\}. \tag{9}$$

Similar to before in (1), we construct a learned model $\bar{T}_Y$, now looking at the abstract next-states that were reached:

$$\bar{T}_Y(\bar{s}' | \bar{s}, a) \triangleq \frac{1}{m} \sum_{i=1}^{m} \mathbb{1}\{\bar{Y}_{\bar{s}, a}^{(i)} = \bar{s}'\}. \tag{10}$$

If this model would be equal, or close, to the transition function $\bar{T}_\omega$ of an abstract MDP $\bar{M}_\omega$, for some valid $\omega$, we could upper bound the loss in performance due to applying learned policy $\bar{\pi}^*$ to $M$ instead of the optimal policy $\pi^*$ [1, 29].

---

[5]The assumption that the reward function is known simplifies our arguments but can be relaxed.

| **Algorithm 1** Procedure: Abstracted MBRL | **Algorithm 2** COLLECTSAMPLES Online |
|---|---|
| **Input:** $M, \phi, \delta, \epsilon, \pi$ | **Input:** $M, \phi, \delta, \epsilon, \pi$ |
| $\bar{Y} = \text{COLLECTSAMPLES}(M, \phi, \delta, \epsilon, \pi)$ | $s = $ initial state |
| The sampling results in sequences $\bar{Y}_{\bar{s},a}$, one for | // The number of samples $m$ is based on the |
| every pair $(\bar{s}, a)$: | *simulator* analysis, Theorem 1. |
| $\bar{Y}_{\bar{s},a} = \phi(s'^{(1)}), \cdots, \phi(s'^{(m)})$ | $\kappa = \delta/(|\bar{S}||A|)$ |
| $= \bar{s}'^{(1)}, \cdots, \bar{s}'^{(m)}$ | $m = \lceil \frac{2[\ln(2^{|\bar{S}|}-2)-\ln(\kappa)]}{\epsilon^2} \rceil$ |
| **for all** $(\bar{s}, a, \bar{s}') \in \bar{S} \times A \times \bar{S}$ **do** | **for all** $\bar{s} \in \bar{S}$ **do** |
| $\quad \bar{T}_Y(\bar{s}'\|\bar{s},a) = \frac{1}{m}\sum_{i=1}^m \mathbb{1}\{\bar{Y}_{\bar{s},a}^{(i)} = \bar{s}'\}$ | $\quad \bar{Y}_{\bar{s},a} = [\,]$ |
| **end for** | **end for** |
| $\bar{M}_Y := \langle \bar{S}, A, \bar{T}_Y, \bar{R}, \gamma \rangle$ | **while** $\min_{(\bar{s},a)} |Y_{\bar{s},a}| < m$ **do** |
| $\pi_Y^* = \text{Value Iteration}(\bar{M}_Y)$ | $\quad \bar{s} = \phi(s)$ |
| Apply $\pi_Y^*$ to $M$ | $\quad a = \pi(\bar{s})$ |
| | $\quad s' = \text{Step}(s, a)$ |
| | $\quad \bar{Y}_{\bar{s},a}.\text{append}(\phi(s'))$ |
| | $\quad s = s'$ |
| | **end while** |
| | **Return:** Return all $\bar{Y}_{\bar{s},a}$ |

Our main question is: do the finite-sample model learning guarantees of MBRL algorithms still hold in the Abstracted RL setting?

## 3.1 Online data collection

In this section we follow the MBRL method from Algorithm 1, collecting samples online using Algorithm 2.[6] Starting from an initial state the agent follows a policy $\pi$. Instead of observing the states $s$, the agent observes abstract states $\bar{s} = \phi(s)$, see Figure 1.

We make two important assumptions in order to make analysis possible. We assume that the MDP is ergodic [25] [7] and that the policy assigns a positive probability to every action in every abstract state. Together this can guarantee that Algorithm 2 can obtain any finite number of samples for every abstract state-action pair within finite time.

Our question is, can we still use Lemma 1 to guarantee that we learn an accurate model?

Since we learn an abstract transition model $\bar{T}_Y$, we want to be able to guarantee that this learned model will be close to the transition model of some abstract MDP. To define this transition model, we first look at how the data is collected.

In the online data collection, a sample in $\bar{Y}_{\bar{s},a}$ is drawn when the agent takes action $a$ when it is in a ground state $s \in \bar{s}$. Specifically the $i$-th abstract $\bar{Y}_{\bar{s},a}^{(i)} = \bar{s}'$ is drawn from (ground) state $X_{\bar{s},a}^{(i)} = s \in \bar{s}$:

$$\bar{Y}_{\bar{s},a}^{(i)} \sim \Pr(\cdot | X_{\bar{s},a}^{(i)} = s, a). \tag{11}$$

Let $X_{\bar{s},a} = (X_{\bar{s},a}^{(i)})_{i=1}^m$ denote the sequence of ground states $s \in \bar{s}$ from which the agent took action $a$. Each ground state gets a weight according to how often it was sampled from, which we formalize with the weighting function $\omega_X$: $\forall_{(\bar{s},a),s \in \bar{s}} \; \omega_X(s,a) \triangleq \frac{1}{m}\sum_{i=1}^m \mathbb{1}\{X_{\bar{s},a}^{(i)} = s\}$. We use $\omega_X$ to define $\bar{T}_{\omega_X}$ analogous to (8):

$$\forall_{(\bar{s},a),\bar{s}'} \; \bar{T}_{\omega_X}(\bar{s}'|\bar{s},a) = \sum_{s \in \bar{s}} \omega_X(s,a) \sum_{s' \in \bar{s}'} T(s'|s,a). \tag{12}$$

---

[6]The order of $m$ in Algorithm 2, the number of samples we want to collect, is based on the analysis of Model-based Interval Estimation (MBIE) [28].

[7]An ergodic, or recurrent, MDP is an MDP where, under every stationary policy, every state is recurrent (i.e., asymptotically every state will be visited infinitely often) [25].

167 We want to have a concentration inequality to provide bounds on the deviation of the learned model
168 $\bar{T}_Y$ from $\bar{T}_{\omega_X}$, we refer to this inequality as the abstract L1 inequality, similar in form to (3):

$$P(|\bar{T}_Y(\cdot|\bar{s}, a) - \bar{T}_{\omega_X}(\cdot|\bar{s}, a)|_1 \ge \epsilon) \le \delta, \tag{13}$$

169 where $\bar{T}_Y(\cdot|\bar{s}, a)$ is defined according to (10) and $\bar{T}_{\omega_X}$ according to (12).

170 If we could directly obtain i.i.d. samples from $\bar{T}_{\omega_X}$ and base our learned model $\bar{T}_Y$ on the obtained
171 samples, then we would be able to show that the abstract L1 inequality holds by applying Lemma 1.
172 Since in this case, we would have $m$ i.i.d. samples per abstract state-action pair, distributed according
173 to $\bar{T}_{\omega_X}(\cdot|\bar{s}, a)$.

174 However, the samples are not guaranteed to be i.i.d. when the agent follows Algorithm 2 to collect
175 the samples. Since every sample $\bar{Y}^{(i)}$ was obtained after taking action $a$ from state $X_{\bar{s},a}^{(i)} = s \in \bar{s}$, as
176 in (11). These can have different distributions if $X_{\bar{s},a}^{(i)} \ne X_{\bar{s},a}^{(j)}$.

177 **Non Identically Distributed** While Lemma 1 assumes i.i.d. random variables, we show that it also
178 holds when the random variables are independent but not (necessarily) identically distributed.

179 **Lemma 3.** *Let* $X_{\bar{s},a} = s_1, \cdots, s_m$ *be a sequence of states* $s \in \bar{s}$ *and let*
180 $\bar{Y}_{\bar{s},a} = \bar{Y}^{(1)}, \bar{Y}^{(2)}, \cdots, \bar{Y}^{(m)}$ *be independent random variables distributed according to*
181 $\Pr(\cdot|s_1, a), \cdots, \Pr(\cdot|s_m, a)$ *(Eqn. 4). Then, for all* $\epsilon > 0$,

$$\Pr(||\bar{T}_Y(\cdot|\bar{s}, a) - \bar{T}_{\omega_X}(\cdot|\bar{s}, a)||_1 \ge \epsilon) \le (2^{|\bar{S}|} - 2)e^{-\frac{1}{2}m\epsilon^2}. \tag{14}$$

182 The proof can be found in Appendix B. It mostly follows the proof by Weissman et al. [30], which uses
183 Hoeffding's inequality [12] and the union bound [6].[8] Lemma 3 shows that the fact that Hoeffding's
184 inequality does not need identically distributed data can be carried over to the setting from Lemma 1.

185 **Independence** We may be tempted to assume the samples are independent, i.e.,

$$\forall_{\bar{s}_1, \cdots, \bar{s}_m \in (\bar{S})^m} \Pr(\bar{Y}_{\bar{s},a}^{(1)} = \bar{s}_1, \cdots, \bar{Y}_{\bar{s},a}^{(m)} = \bar{s}_m) = \Pr(\bar{Y}_{\bar{s},a}^{(1)} = \bar{s}_1) \cdots P(\bar{Y}_{\bar{s},a}^{(m)} = \bar{s}_m) \tag{15}$$

186 however, this may not be the case:

187 **Observation 1.** *When collecting samples online, i.e., based on Algorithm 2, the samples cannot be*
188 *assumed to be independent.*

189 The following counterexample illustrates this.

190 **Counterexample** To show that the samples may not be indepen-
191 dent, we will give a counterexample. We use the example MDP and
192 abstraction in Figure 2, where we have 4 (ground) states, 3 abstract
193 states and only 1 action. We look at the transition probability from
194 abstract state $A$, $\bar{T}_Y(\cdot|A)$.

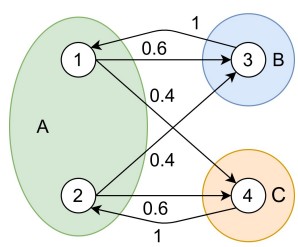

195 We will consider two samples and show that for at least one com-
196 bination of $\bar{s}_1$ and $\bar{s}_2$ the samples are not independent. Consider
197 $\bar{s}_1 = \bar{s}_2 = B$. That is, the first two times that we experience a
198 transition from the abstract state $A$, we end up in $B$.

199 Let state 1 be the starting state. Then we have $\Pr(\bar{Y}_A^{(1)} = B) =$
200 $\Pr(B|1) = 0.6$ and

Figure 2: Simple MDP, with
only 1 action, and abstraction.
The small circles are ground
states (1,2,3,4). A, B and C are
the abstract states. The num-
bers along the arrows show
the transition probabilities, e.g.
$P(3|1) = 0.6$.

$$\Pr(\bar{Y}_A^{(2)} = B) = \sum_{\bar{s} \in \bar{S}} \Pr(\bar{Y}_A^{(2)} = B|\bar{Y}_A^{(1)} = \bar{s}) \Pr(\bar{Y}_A^{(1)} = \bar{s}) \tag{16}$$

$$= 0 + 0.6 \cdot 0.6 + 0.4 \cdot 0.4 = 0.52. \tag{17}$$

201 So then we end up with: $\Pr(\bar{Y}_A^{(1)} = B) \Pr(\bar{Y}_A^{(2)} = B) = 0.6 \cdot 0.52 = 0.321$. And for the joint
202 probability: $\Pr(\bar{Y}_A^{(1)} = B, \bar{Y}_A^{(2)} = B) = \Pr(\bar{Y}_A^{(1)} = B) \Pr(\bar{Y}_A^{(2)} = B|\bar{Y}_A^{(1)} = B) = 0.6 \cdot 0.6 =$
203 $0.36$.

---

[8]Hoeffding's inequality and the union bound can be found in Appendix A.

Thus we have that $\Pr(\bar{Y}_A^{(1)} = B, \bar{Y}_A^{(2)} = B) \neq \Pr(\bar{Y}_A^{(1)} = B)\Pr(\bar{Y}_A^{(2)} = B)$, the samples are not independent. Leading us to the second observation:

**Observation 2.** *As independence cannot be guaranteed, Lemmas 1 and 3 cannot be readily applied to show that the abstract L1 inequality holds.*

### 3.2 Simulator data collection

Here we also want to give a guarantee in the form of the abstract L1 inequality from (13). While in the previous section we found this was not possible because the samples were dependent, here we assume that we have access to a simulator. To some extent this is not surprising, but to the best of our knowledge, this is the first work that explicitly shows how to combine MBRL and abstraction, using a simulator. We assume that this allows us to select (or move to) any state and draw a sample from its transition function. This we call the independent samples assumption:

**Assumption 1** (Independent samples)**.** *We assume we can obtain independent samples, e.g. for any state-action pair $(s, a)$ we can draw samples directly from its transition function $T(\cdot|s, a)$.*

In case a simulator of the MDP is available this is a reasonable assumption. For every $(\bar{s}, a)$ the simulator sampling procedure (Algorithm 3 in Appendix B) selects a prototype $x_{\bar{s},a} \in \bar{s}$ to sample from. We define a weighting function $\omega_x(s, a)$ that has weight 1 if $s$ is the prototype $x_{\bar{s},a}$ and 0 otherwise:

$$\forall_{(\bar{s},a), s \in \bar{s}} \; \omega_x(s, a) \triangleq \mathbb{1}\{s = x_{\bar{s},a}\}. \tag{18}$$

Then we use this $\omega_x$ to define the abstract transition function $\bar{T}_{\omega_X}$ according to (8). $\bar{T}_{\omega_x}(\bar{s}'|\bar{s}, a) = \sum_{s' \in \bar{s}'} T(s'|s = x_{\bar{s},a}, a)$. This way the samples that we collect for one pair $(\bar{s}, a)$ are i.i.d., they are independent because of our assumption of independent samples and identically distributed because we sample from the prototype. This means we can use Lemma 1. We show that with the simulator we can combine MBRL and abstraction, and still learn an accurate model, that is, we can guarantee that $\bar{T}_Y$ will be close to $\bar{T}_{\omega_x}$, with high probability:

**Theorem 1.** *Under assumption 1, and following the procedure in Algorithm 1, with the data collection from Algorithm 3 (Appendix B), with inputs $|\bar{S}|, A, \epsilon$ and $\delta$. For $\bar{T}_Y$ constructed by the algorithm we have that with probability $1 - \delta$, the following holds:*

$$\forall_{(\bar{s},a)} \; ||\bar{T}_Y(\cdot|\bar{s}, a) - \bar{T}_{\omega_x}(\cdot|\bar{s}, a)||_1 \leq \epsilon. \tag{19}$$

By Assumption 1 we can obtain any number of independent samples for each abstract state action pair $(\bar{s}, a)$. Using Lemma 1 we can then derive the number of samples $m$ that is required for each pair $(\bar{s}, a)$ such that, after applying a union bound, we obtain the bounds in (19). The full proof can be found in Appendix B.

## 4 Related work

There is a lot of work that considers the combination of abstraction with either planning or (online) RL. In a lot of these works the dependence of samples that arises in Abstracted RL is not an issue due to various assumptions, similarly to how in MBRL dependence of samples is often not an issue because of the Markov property and the assumption that the MDP is communicating [25]. Often this is either due the assumption that data has been obtained i.i.d., the specific type of abstraction, or because access to an MDP model is assumed.

One paper that does give a result for the Abstracted RL setting is the work by Abel et al. [2]. They show that in this setting R-MAX [7] no longer maintains its guarantees when paired with any type of state-abstraction function, though their example is specifically for approximate Q-function abstractions. They also show that the expected trajectory of a learning agent in a constructed abstract MDP (Definition 3) is not the same as in Abstracted RL. Their work makes clear there is a complication when combining MBRL and abstraction, here we further investigated the cause of this complication, the dependence between samples.

For planning in constructed abstract MDPs, some main results for exact state-abstractions come from Li et al. [18] and for approximate state-abstractions from Abel et al. [1]. The results from Abel et al. [1] allow for quantifying an upper bound on performance for policies found in a constructed

abstract MDP, as in section 2.2. Taïga et al. [29] build on this by giving a result for performing RL on top of the constructed abstract MDP. They provide upper bounds for this setting when using MBIE with exploratory bonus (MBIE-EB) [28]. In addition, they give an example to show that in this combination you cannot guarantee optimal performance in the original MDP. Still, they show that an upper bound on the loss in value can be given.

Both Paduraru et al. [24] and Jiang et al. [15] deal with the issue of dependence by making the explicit assumption that samples are obtained i.i.d. Paduraru et al. [24] consider the setting where we are given a dataset for a continuous domain and then use discretization to aggregate states into abstract states. They then give PAC-style guarantees on the learned abstract model and the value that a policy based on this model can achieve in the real MDP. Instead of using the L1 deviation bound from Weissman et al. [30], Paduraru et al. [24] use a similar bound for i.i.d. samples by Devroye and Gyorfi [9], which requires a minimum amount of samples. Another difference is that their results calculate the probability that the model will be $\epsilon$-accurate given a fixed dataset. They assume that the data has been gathered i.i.d., but our Lemma 3 shows that merely independent data would be enough. At the same time, our results show that when we collect data online in the Abstracted RL setting, their guarantees will not hold.

Jiang et al. [15] operate in the abstraction selection setting, where the agent is provided with a set of abstraction functions (state representations). They do not assume that any of the abstraction functions results in a Markov model, but they do assume a given dataset, with data that was collected i.i.d. They give a bound directly on how accurate the Q-values based on the (implicitly) learned model will be, rather than on the accuracy of the model itself. As we showed, the assumption that the data is i.i.d. is not a trivial assumption, since it means the data cannot just have been collected online. They do mention that samples will not be strictly independent if a fixed exploration policy is used to collect data but do not mention what the implications are.

There are quite a few other papers in the abstraction selection setting, several of these assume that the given set of state representations contains a Markov model [11, 19, 23]. Hallak et al. [11] give asymptotic guarantees for selecting the correct model and on building an exact MDP model. The assumption that there is an MDP model in the given set of representations is crucial in their analysis since for this 'true model' the samples are i.i.d. Similarly, both Maillard et al. [19] and Ortner et al. [23] also assume that the given set of state representations contains a Markov model. They create an algorithm for which they obtain regret bounds, their analysis also makes use of the Markov representation.

Other work in the abstraction selection setting does not assume that the set of abstraction functions contains a Markov model [16, 22]. However, Ortner et al. [22] use Theorem 2.1 from Weissman et al. [30] that requires i.i.d. samples, which we have shown here cannot be guaranteed in this setting. Lattimore et al. [16] operate in a setting more general than MDPs, where the dynamics of the true environment depend arbitrarily on a history of actions, rewards, and observations. The agent gets as input a finite set of environments, one of which is the true environment. Since the input includes the full model of each environment, the agent does not have to learn a transition model. Instead, to obtain regret bounds, they directly compare the rewards the agent obtains to the expected rewards of the given environments and eliminate environments that are implausible given the observed rewards.

Another way to deal with the issue of dependence is by looking at convergence in the limit [27, 13, 20]. Singh et al. [27] give an asymptotic result for the convergence of Q-learning and TD(0) in MDPs with soft state aggregation. Soft state aggregation means that a state $s$ belongs to a cluster $x$ with some probability $P(x|s)$, this means a state $s$ can belong to several clusters. The state-abstraction functions we consider are a special case of this, where each state is part of exactly one abstract state (or cluster). Their result relies on having a stationary policy that assigns a non-zero probability to every action in every state and the assumption that the MDP is ergodic. Together these imply there is a limiting state distribution, and using this they show convergence asymptotically. Our main interest is in finite-samples guarantees with policies that change due to exploration, whereas this work gives convergence guarantees in the limit using a fixed policy.

Hutter [13] gives a variety of results focusing on both approximate and exact abstractions in environments without MDP assumptions. Several of these are in the planning setting, similar to those of Abel et al. [1]. Most relevant for us is their Theorem 12, which for online RL shows convergence in the limit of the empirical transition function under weak conditions, e.g. if the abstract process itself

is an MDP. Under this condition however the problem reduces to RL in an (abstract) MDP, rather than Abstracted RL.

Majeed and Hutter [20] build on the work by Hutter [13] and focus on the combination of model-free RL and exact abstraction. They show that, under the condition of state uniformity, q-learning can be shown to converge in the limit to the optimal solution. State uniformity means that histories that are grouped together have the same optimal q-values. In contrast to our setting, they look at an exact abstraction, extending it to approximate aggregation was left as an open question.

Other related work is in the area of MDPs with rich observations or block structure [4, 10]. However, in that setting each observation can be generated only from a *single* hidden state, which means that the issue of non-i.i.d. data due to abstraction does not arise. In contrast, in our setting multiple (hidden) states generate the same observation. Azizzadenesheli et al. [4] state their setting can be seen as an aggregation problem, where the observations can be aggregated to form a small (latent) MDP. But in our case, we do not try to learn the MDP (as it is not small). Du et al. [10] describe that their setting is similar to exact model similarity (or bisimulation), but we focus on approximate model similarity which is what introduces the problems as described here.

# 5    Discussion

When collecting samples online in Abstracted RL, there is a potential dependence between samples, meaning we cannot use the typically used concentration results that assume i.i.d. samples, e.g. Theorem 2.1 from Weissman et al. [30], the empirical Bernstein inequality [3, 21] or the Chernoff bound. In case the samples are only weakly dependent, it may be that concentration inequalities for (weakly) dependent variables are a viable alternative through which we can come to guarantees on the learned model. Alternatively, it may be possible to change the sampling process to ensure independent samples. One way to ensure independent samples is to, as in the simulator setting, select a prototype state and only use the samples collected from this state. Though in this case, we will be discarding information when we reach a state $s \in \bar{s}$ that is not the prototype.

Our assumption on the simulator that we can go/reset to any state to draw samples from it can be relaxed, though it may mean that the procedure takes considerably more time. Consider the case where we cannot just reset the simulator to the state $s$ from which we want to sample, and instead, it would behave like the MDP. In this case, we would have to take the right actions to arrive at the state $s$ from which we would like to sample. Since we assume we do not know $T$, this may take a long time. This also shows the difficulty of assuming that in the MBRL setting somehow have access to an i.i.d. dataset, as has been assumed in some earlier work [24, 15].

# 6    Conclusion

We analyzed Abstracted RL: the combination of MBRL and state abstraction when the model of the MDP is not available. We have shown that in Abstracted RL samples obtained online cannot be assumed to be independent. Since many current guarantees from MBRL methods rely on this assumption, their guarantees do not hold in this setting. And in fact, no current methods exist that give (correct) finite-sample quality guarantees for the models learned in this setting. This also means that current results that rely on an i.i.d. assumption cannot be readily transferred to the Abstracted RL setting.

In addition, we show that with a simulator, since we can draw independent samples, it is still possible to give guarantees on the accuracy of the model. However, having access to a simulator may often not be possible. An important step is to see if the MBRL guarantees can be adapted to Abstracted RL for online sample collection.

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
