# An Analysis of Abstracted Model-Based Reinforcement Learning, Appendix

## A  Well known results

2

### A.1  Hoeffding's Inequality

Hoeffding's inequality can tell us what the probability is that the average of $m$ random independent (but not necessarily identically distributed) samples deviates more than $\epsilon$ from its expectation.

Let $Z^{(1)}, Z^{(2)}, \cdots, Z^{(m)}$ be bounded independent random variables and let $\bar{Z}$ and $\mu$ be defined as:

$$\bar{Z} \triangleq \frac{Z^{(1)} + \cdots + Z^{(m)}}{m}, \tag{1}$$

$$\mu \triangleq E[\bar{Z}] = \frac{E[Z^{(1)} + \cdots + Z^{(m)}]}{m}. \tag{2}$$

Then Hoeffding's inequality states:

**Lemma 1** (Hoeffding's inequality [2]). *If $Z^{(1)}, Z^{(2)}, \cdots, Z^{(m)}$ are independent and $0 \leq Z^{(i)} \leq 1$ for $i = 1, \cdots, m$, then for $0 < \epsilon < 1 - \mu$*

$$\Pr(\bar{Z} - \mu \geq \epsilon) \leq e^{-2m\epsilon^2}$$

### A.2  Union Bound

Given that we have a set of events, the union bounds allows us to upper bound the probability that at least one of the events happens, even when these events are not independent.

**Lemma 2** (Union Bound [1]). *For a countable set of events $A_1, A_2, A_3, \cdots$, we have*

$$\Pr(\cup_i A_i) \leq \sum_i \Pr(A_i). \tag{3}$$

I.e., the probability that at least one of the events happens is at most the sum of the probabilities of the individual events.

## B  Proofs

15

Submitted to 35th Conference on Neural Information Processing Systems (NeurIPS 2021). Do not distribute.

## B.1 Proof of Lemma 3

*Proof.* The proof mostly follows the steps by Weissman et al. [4].

To shorten notation we define $P_Y \triangleq \bar{T}_Y(\cdot|\bar{s}, a)$ and $P_{\omega_X} \triangleq \bar{T}_{\omega_X}(\cdot|\bar{s}, a)$.

We will make use of the following result, shown in Levin and Peres [3] (proposition 4.2), that for any distribution $Q$ on $\bar{S}$

$$||Q - P_{\omega_X}||_1 = 2 \max_{\bar{\mathcal{S}} \subseteq \bar{S}}(Q(\bar{\mathcal{S}}) - P_{\omega_X}(\bar{\mathcal{S}}),$$

where $\bar{\mathcal{S}}$ is a subset of $\bar{S}$ and $P_{\omega_X}(\bar{\mathcal{S}}) = \sum_{\bar{s}' \in \bar{\mathcal{S}}} P_{\omega_X}(\bar{s}')$. Thus we have that

$$||P_Y - P_{\omega_X}||_1 = 2 \max_{\bar{\mathcal{S}} \subseteq \bar{S}}(P_Y(\bar{\mathcal{S}}) - P_{\omega_X}(\bar{\mathcal{S}})). \tag{4}$$

Using this we can write

$$\Pr(||P_Y - P_{\omega_X}||_1 \geq \epsilon) = \Pr\left[2 \max_{\bar{\mathcal{S}} \subseteq \bar{S}}\left[P_Y(\bar{\mathcal{S}}) - P_{\omega_X}(\bar{\mathcal{S}})\right] \geq \epsilon\right] \tag{5}$$

$$= \Pr\left[\max_{\bar{\mathcal{S}} \subseteq \bar{S}}\left[P_Y(\bar{\mathcal{S}}) - P_{\omega_X}(\bar{\mathcal{S}})\right] \geq \frac{\epsilon}{2}\right] \tag{6}$$

$$= \Pr\left[\cup_{\bar{\mathcal{S}} \subseteq \bar{S}}\left[P_Y(\bar{\mathcal{S}}) - P_{\omega_X}(\bar{\mathcal{S}}) \geq \frac{\epsilon}{2}\right]\right] \tag{7}$$

$$\leq \sum_{\bar{\mathcal{S}} \subseteq \bar{S}} \Pr\left[P_Y(\bar{\mathcal{S}}) - P_{\omega_X}(\bar{\mathcal{S}}) \geq \frac{\epsilon}{2}\right], \tag{8}$$

where the last step follows from the union bound.

Assuming $\epsilon > 0$, we have for $\bar{\mathcal{S}} = \bar{S}$ and for $\bar{\mathcal{S}} = \emptyset$ that $\Pr(P_Y(\bar{\mathcal{S}}) - P_{\omega_X}(\bar{\mathcal{S}}) \geq \frac{\epsilon}{2}) = 0$.

For every other subset $\bar{\mathcal{S}}$, we can define a random binary variable that is 1 when $Y^{(i)} \in \bar{\mathcal{S}}$ and 0 otherwise. We have that $P_{\omega_X}(\bar{\mathcal{S}})$ acts as $\mu$ (2) from Lemma 1 and $P_Y(\bar{\mathcal{S}})$ as $\bar{Z}$ (1). Thus applying Lemma 1 to this random variable we have:

$$\Pr(P_Y(\bar{\mathcal{S}}) - P_{\omega_X}(\bar{\mathcal{S}}) \geq \frac{\epsilon}{2}) \leq e^{-2m\frac{\epsilon}{2}^2} = e^{-\frac{1}{2}m\epsilon^2}. \tag{9}$$

Then it follows that

$$\Pr(||P_Y - P_{\omega_X}||_1 \geq \epsilon) \leq \sum_{\bar{\mathcal{S}} \subseteq \bar{S}} \Pr(P_Y(\bar{\mathcal{S}}) - P_{\omega_X}(\bar{\mathcal{S}}) \geq \frac{\epsilon}{2}) \tag{10}$$

$$\leq \sum_{\bar{\mathcal{S}} \subset \bar{S}: \bar{\mathcal{S}} \neq \bar{S}, \emptyset} \Pr(P_Y(\bar{\mathcal{S}}) - P_{\omega_X}(\bar{\mathcal{S}}) \geq \frac{\epsilon}{2}) \tag{11}$$

$$\leq (2^{|\bar{S}|} - 2)e^{-\frac{1}{2}m\epsilon^2}, \tag{12}$$

where $\bar{S} \subset \bar{\mathcal{S}} : \bar{S} \neq \bar{\mathcal{S}}, \emptyset$ denotes that the empty set $\emptyset$ and the full set $\bar{\mathcal{S}}$ are excluded. $\qquad \square$

## B.2 Simulator Setting, proof of Theorem 1

Before starting with the actual proof, we first shortly go over Algorithm 3 and give two lemmas that the proof uses.

The agent will draw samples using the simulator as described in Algorithm 3. Since we assume that we can sample directly from the transition functions $T(\cdot|s, a)$, this algorithm just loops over all pairs $(\bar{s}, a)$ and samples $m$ times[1] from each transition function. More formally, for every pair $(\bar{s}, a)$ the algorithm selects one prototype state $x_{\bar{s},a} = s \in \bar{s}$. Then, it loops over every pair $(\bar{s}, a)$ and samples $m$ transitions from $T(\cdot|x_{\bar{s},a}, a)$. The set of collected experiences for each abstract state-action pair $(\bar{s}, a)$ is represented by $\bar{Y}_{\bar{s},a}$, as defined by (9).

---

[1]The value of $m$ in Algorithm 3 is chosen based on the results further along in this section.

**Algorithm 3** COLLECTSAMPLES with Simulator

---

**Input:** $M, \phi, \delta, \epsilon$
$\kappa = \frac{\delta}{|\bar{S}||A|}$
$m = \lceil \frac{2[\ln(2^{|\bar{S}|}-2)-\ln(\kappa)]}{\epsilon^2} \rceil$
**for all** $(\bar{s}, a) \in \bar{S} \times A$ **do**
  $\bar{Y}_{\bar{s},a} = [\,]$
  $x_{\bar{s},a} = $ select a prototype state $s \in \bar{s}$
  **for** $i = 1 : m$ **do**
    $s' = \text{Sample}(T(\cdot | x_{\bar{s},a}, a))$
    $\bar{Y}_{\bar{s},a}.\text{append}(\phi(s'))$
  **end for**
**end for**
**Return:** all $\bar{Y}_{\bar{s},a}$

---

Given $\bar{Y}_{\bar{s},a}$, the learned model $\bar{T}_Y(\cdot|\bar{s}, a)$ is defined according to (10) and $\bar{T}_{\omega_x}$ is defined according to (8), with $\omega_x$ defined according to (18). It follows from Lemma 1 that we can derive a number of samples that we require to guarantee that, for inputs $\kappa$ and $\epsilon$, $\Pr(||\bar{T}_Y(\cdot|\bar{s}, a) - \bar{T}_{\omega_x}(\cdot|\bar{s}, a)||_1 \geq \epsilon) \leq \kappa$ is true:

**Lemma 4.** *For inputs $\kappa$ and $\epsilon$ $(0 < \kappa < 1, 0 < \epsilon < 2)$, we have that for $m \geq \frac{2[\ln(2^{|\bar{S}|}-2)-\ln(\kappa)]}{\epsilon^2}$ the following holds:*

$$\Pr(||\bar{T}_Y(\cdot|\bar{s}, a) - \bar{T}_{\omega_x}(\cdot|\bar{s}, a)||_1 \geq \epsilon) \leq \kappa. \tag{13}$$

*Proof.* To shorten notation we again use the definitions $P_Y \triangleq \bar{T}_Y(\cdot|\bar{s}, a)$ and $P_{\omega_x} \triangleq \bar{T}_{\omega_x}(\cdot|\bar{s}, a)$. We have from Lemma 1 that

$$\Pr(||P_Y - P_{\omega_x}||_1 \geq \epsilon) \leq (2^{|\bar{S}|} - 2)e^{-\frac{1}{2}m\epsilon^2}. \tag{14}$$

We need to select $m$ such that $\kappa \geq (2^{|\bar{S}|} - 2)e^{-\frac{1}{2}m\epsilon^2}$:

$$\kappa \geq (2^{|\bar{S}|} - 2)e^{-\frac{1}{2}m\epsilon^2} \tag{15}$$

$$\frac{\kappa}{2^{|\bar{S}|} - 2} \geq e^{-\frac{1}{2}m\epsilon^2} \tag{16}$$

$$\ln(\kappa) - \ln(2^{|\bar{S}|} - 2) \geq -\frac{m\epsilon^2}{2} \tag{17}$$

$$\frac{m\epsilon^2}{2} \geq \ln(2^{|\bar{S}|} - 2) - \ln(\kappa) \tag{18}$$

$$m \geq \frac{2[\ln(2^{|\bar{S}|} - 2) - \ln(\kappa)]}{\epsilon^2} \tag{19}$$

Thus if $m \geq \frac{2[\ln(2^{|\bar{S}|}-2)-\ln(\kappa)]}{\epsilon^2}$ we have

$$\Pr(||P_Y - P_{\omega_x}||_1 \geq \epsilon) \leq \kappa. \qquad \square$$

Using the Union bound, we can give a lower bound on the probability that, for every $(\bar{s}, a)$, $\bar{T}_Y(\cdot|\bar{s}, a)$ and $\bar{T}_{\omega_x}(\cdot|\bar{s}, a)$ are $\epsilon$ close:

**Lemma 5.** *If*

$$\forall_{(\bar{s},a)} \left[ \Pr(||\bar{T}_Y(\cdot|\bar{s}, a) - \bar{T}_{\omega_x}(\cdot|\bar{s}, a)||_1 \geq \epsilon) \right] \leq \frac{\delta}{|\bar{S}||A|} \tag{20}$$

*then with probability at least $1 - \delta$ the following holds:*

$$\max_{(\bar{s},a)} \left[ ||\bar{T}_Y(\cdot|\bar{s}, a) - \bar{T}_{\omega_x}(\cdot|\bar{s}, a)||_1 \right] \leq \epsilon. \tag{21}$$

*Proof.* We define

$$\Delta_{\bar{s},a} \triangleq ||\bar{T}_Y(\cdot|\bar{s},a) - \bar{T}_{\omega_x}(\cdot|\bar{s},a)||_1. \tag{22}$$

Then $\Pr(\max_{(\bar{s},a)}\{\Delta_{\bar{s},a} \geq \epsilon\})$ is the probability that for at least one abstract state-action pair $\Delta_{\bar{s},a} \geq \epsilon$. From the union bound it follows that $\Pr(\max_{(\bar{s},a)}\{\Delta_{\bar{s},a} \geq \epsilon\}) \leq \delta$:

$$\Pr(\max_{(\bar{s},a)}\{\Delta_{\bar{s},a} \geq \epsilon\}) \leq \sum_{\bar{s},a} \Pr(\Delta_{\bar{s},a} \geq \epsilon) \tag{23}$$

$$\leq \sum_{\bar{s},a} \frac{\delta}{|\bar{S}||A|} \tag{24}$$

$$= \delta. \tag{25}$$

Since $\Pr(\max_{(\bar{s},a)}\{\Delta_{\bar{s},a} \leq \epsilon\}) = 1 - \Pr(\max_{(\bar{s},a)}\{\Delta_{\bar{s},a} \geq \epsilon\})$ it follows that $\Pr(\max_{(\bar{s},a)}\{\Delta_{\bar{s},a} \leq \epsilon\}) \geq 1 - \delta$. Thus the probability that (21) holds is at least $1 - \delta$. $\qquad\square$

Now we are ready to proof Theorem 1:

*Proof of Theorem 1.* By Assumption 1, and the earlier assumption that $|S|$ and $|A|$ are finite, we have that for every abstract state-action pair we can obtain $m$ samples, for any $m > 0$, in finite time. Given the inputs $|\bar{S}|$, $A$, $\epsilon$ and $\delta$, Algorithm 3 sets $m = \lceil \frac{2[\ln(2^{|\bar{S}|}-2)-\ln(\kappa)]}{\epsilon^2} \rceil$, where $\kappa = \frac{\delta}{|\bar{S}||A|}$. Then for every $(\bar{s},a)$ a prototype state $x_{\bar{s},a} = s \in \bar{s}$ is selected. We use (18) to define $\omega_x$ and (8) to define $\bar{T}_{\omega_x}$.

For all $(\bar{s},a)$ Algorithm 3 obtains a sequence $\bar{Y}_{\bar{s},a}$ by sampling from the transition function from the prototype state $x_{\bar{s},a}$ and Algorithm 1 constructs the empirical transition functions as in (10).

Given our choice of $m$ it follows from Lemma 4, with inputs $\kappa = \frac{\delta}{|\bar{S}||A|}$ and $\epsilon$, it holds that

$$\forall_{(\bar{s},a)} \ \Pr(||\bar{T}_Y(\cdot|\bar{s},a) - \bar{T}_{\omega_x}(\cdot|\bar{s},a)||_1 \geq \epsilon) \leq \frac{\delta}{|\bar{S}||A|}. \tag{26}$$

Then by Lemma 5 we have that, with probability at least $1 - \delta$, (19) holds. $\qquad\square$