# OpenReview forum: "An Analysis of Abstracted Model-Based Reinforcement Learning"
_NeurIPS.cc/2021/Conference — NeurIPS 2021 Submitted_

### Official Review · Reviewer_9nCd · 2021-06-27

**Rating:** 5
**Confidence:** 2

**Summary:**

In this paper, the transition model accuracy guarantee in model-based RL on abstract (tabular) MDP is discussed.

The problem that the sample independence assumption, which is important for guaranteeing meaningful model accuracy bound, is not always valid when the state is abstracted by approximate model similarity abstraction.

To address this problem (to realize i.i.d sampling),
the use of a simulator, which can re-start from arbitral state-action pair, is proposed.


**Limitations And Societal Impact:**

yes

**Main Review:**

Key strength:
The key insight "the simulator in which the agent can restart from any state-action pair" is interesting.
In the general RL setting (even if the simulators are used), the agent collects trajectory starting from the initial states, and the use of the "restart" function is not focused on.
The insight of this paper promotes utilizing the restart function.
(For now, the insight is for MBRL setting on certain abstract MDP, but the insight of the paper motivates post cursors to conduct a discussion about the benefit of the simulator equipped with the restart function.)

Key weakness:
No empirical result is provided. I understand the paper is a theoretical one, but the quality of the paper should be much improved if the empirical result (even in a simple grid-world case) to support the authors claim or insight is provided (e.g., is Algorithm 3 actually works well?).

Typo:
Line 374: rl -> RL
Line 397: q-learning -> Q-leanring


**Time Spent Reviewing:**

8

---

> ### Author Response · Authors · 2021-08-09
> **Initial author response**
>
> Thank you for your analysis of our work, we are grateful for the feedback.
>
> “No empirical result is provided. I understand the paper is a theoretical one, but the quality of the paper should be much improved if the empirical result (even in a simple grid-world case) to support the authors claim or insight is provided (e.g., is Algorithm 3 actually works well?).”
>
> In this case, we think our main contribution is a theoretical one. Our main contribution is pointing out a problem in the combination of MBRL and abstraction, that has gone largely unaddressed in the literature. The appropriate tool to clarify this is using theory.
>
> We point out that the question in how far this problem occurs ‘in practice’ could be interesting, but would be very difficult to answer: what is the appropriate set of benchmarks to investigate this? What abstractions should be considered? How much data would need to be collected? Using what policy? What would even be the metric of success? etc. As such, such an empirical study is well beyond the scope of the current paper.

---

### Official Review · Reviewer_qhw2 · 2021-07-16

**Rating:** 7
**Confidence:** 4

**Summary:**

This paper deals with model learning when given access to a specific type of abstraction, model-irrelevance (or bisimulation). It states the when the data is collected online, learning an accurate model in the abstract MDP is not possible since the samples are no longer independent (as required by MBRL methods that use concentration inequalities to bound the error of the estimated model). It then shows that given simulator access, this problem no longer exists as samples can be collected from anywhere in the state space.

**Ethical Concerns:**

I do not think there any explicit ethical concerns raised by this work.

**Limitations And Societal Impact:**

Suggestions: My suggestion would be to bend the main problem the authors try to answer here. This is again pointed to in my comments in the 'originality' section. I think doing that can push this paper towards a clear acceptance.

I do not think there any explicit negative societal impacts of this work.


**Main Review:**

Originality: I really like the motivation of this work, but I think the writing eventually ends up confusing the reader more than highlighting the main problem. To me, the motivation seems to be to show why/when model learning guarantees can fail when using function approximation (a general state abstraction technique). That, I think is a good question to ask. And it in the context of independence of samples, it would then lead to the following question: Even in the case of iid data, can function approximation (or state abstraction) induce dependence in the samples and thus rule out guarantees for methods that learn a model in the abstracted space? However, the authors stick to the case of online data collection, which in and of itself leads to dependence in samples. Therefore, I believe the working of the paper from that point on does not shed much insight. In other words, now irrespective of whether we are in the original MDP or in the abstract MDP, the samples are dependent in both cases. I don't think the authors' motivation stems from this setting. Rather, the setting should be about the dependence in samples induced by abstraction even when the data is iid.

Quality and Clarity: Unfortunately, the paper lacks massively in these two regards. It is hard to follow, sometimes due to bad/nonintuitive notation and sometimes due to sloppy writing. I think improving in this regard can help bring out the right motivation of the paper, which I think holds good value.

Significance: In it's current state, I do not think the paper is significant enough. However, if the authors can clarify the motivation, improve the writing, and slightly change the main question they try to ask (as detailed above), I think the paper and the negative result could be extremely valuable to both the theoretical and empirical RL communities.

Questions/Clarifications:

My main question is regarding the following statement + on the above comments:

"Other related work is in the area of MDPs with rich observations or block structure [4, 10]. However, in that setting each observation can be generated only from a single hidden state, which means that the issue of non-i.i.d. data due to abstraction does not arise."

It seems to me that here the authors are trying to say that the abstraction property is key to dependence in samples. However, I cannot gather what exactly they are referring to. Why is the state abstraction non-stationary? Why do the authors call the abstract MDP as a POMDP? What is so special about bisimulation that results in dependence? Looking at the counter examples given in the paper, it seems like the dependence is simply because of the online data collection procedure, and not due to the state abstraction itself. I think clarifying this really holds the key to the contributions of this paper.


--------------- Score updated from 4 to 7 after multiple discussions with the authors ---------------

**Time Spent Reviewing:**

6 hrs

---

> ### Author Response · Authors · 2021-08-09
> **Initial author response**
>
> Thank you for your analysis and the clearly structured feedback, it is greatly appreciated.
>
> We think one of the questions points to a source of a number of misconceptions, and therefore treat this first:
> “ “Other related work is in the area of MDPs with rich observations or block structure [4, 10]. However, in that setting each observation can be generated only from a single hidden state, which means that the issue of non-i.i.d. data due to abstraction does not arise."
>
> It seems to me that here the authors are trying to say that the abstraction property is key to dependence in samples. However, I cannot gather what exactly they are referring to.”
>
> In the block mdp setting, one (Markov) state can omit multiple observations. But each observation can only come from one (Markov) state. In this setting you can just map each observation to its corresponding (Markov) state. In our setting the opposite is true. Multiple (Markov) states map to a single observation (an abstract state), so an abstract state can not directly be mapped to a single (Markov) state. This type of abstraction is a realistic setting in the literature, see e.g. Abel et al. (2016), Near Optimal Behavior via Approximate State Abstraction.
>
> “Even in the case of iid data, can function approximation (or state abstraction) induce dependence in the samples and thus rule out guarantees for methods that learn a model in the abstracted space?”
>
> Indeed, function approximation can be seen as a special case of state abstraction, where there are classes of states that are mapped to the same values.
>
> We show that as long as we can sample underlying states "from each abstract state" independently (using a simulator for instance) then things work out OK. So in fact, our paper gives an answer to the question posed by the reviewer: if underlying states are drawn independently, function approximation will not lead to a problem.
>
> “However, the authors stick to the case of online data collection, which in and of itself leads to dependence in samples. ... In other words, now irrespective of whether we are in the original MDP or in the abstract MDP, the samples are dependent in both cases. I don't think the authors' motivation stems from this setting.”
>
> “Looking at the counter examples given in the paper, it seems like the dependence is simply because of the online data collection procedure, and not due to the state abstraction itself.”
>
> We disagree with these remarks, what we show in the counterexample is specifically that adding abstraction is what changes the samples to be dependent. In the original MDP, assuming ergodicity, the next states s' for each s,a are independent.
>
> If, in our counter example, we would only look at the probability that the first x transitions from state 1 result in state 3 we would see that P(Y_1 = 3, Y_2 = 3, … Y_x = 3) = P(Y_1 = 3) * P(Y_2 = 3) …* P(Y_x = 3).
>
> What we show in the counterexample is specifically that adding abstraction is what causes the samples to be dependent.
>
> “Why is the state abstraction non-stationary? “
> We interpret this question as “why does abstraction lead to states that are non-stationary” (the abstraction itself does not change over time). That is because the true transition probabilities depend on the underlying state, not on the abstracted states.
>
> “Why do the authors call the abstract MDP as a POMDP?”
> We call the abstract MDP a POMDP because, as illustrated in Figure 1, we assume the agent acts in a (ground) MDP but only sees in which abstract state it is, rather than the true state. This can be seen as a POMDP with deterministic observations, where the observation function returns the abstract state. This explicit link between state abstraction and a POMDP that we make has been under-appreciated in the literature (it has been made before, e.g. by Bai et al. (2016), Markovian State and Action Abstractions for MDPs via Hierarchical MCTS.), but we feel it is a very helpful perspective in understanding the problem that we identify: for POMDPs it is well-known that acting based on just the last observation can lead to poor performance, which essentially is what our considered form of state abstraction does. Here we additionally show it can lead to complications when trying to use existing model-based RL techniques.
>
> “It is hard to follow, sometimes due to bad/nonintuitive notation and sometimes due to sloppy writing.”
> Could you give us some specific points to improve, e.g. what is some of the non-intuitive notation?

---

> > ### Comment · Reviewer_qhw2 · 2021-08-17
> > **Re: Initial author response**
> >
> > *"In the block mdp setting, one (Markov) state can omit multiple observations."* + *"We call the abstract MDP a POMDP because, as illustrated in Figure 1, we assume the agent acts in a (ground) MDP but only sees in which abstract state it is, rather than the true state."*
> >
> > - I see the connection you are trying to make now. This looks interesting but I would argue that this is only so if the abstraction is not Markovian in nature. I do not think that is the case for bisimulation though, i.e. it is Markovian.
> >
> > *"We show that as long as we can sample underlying states "from each abstract state" independently (using a simulator for instance) then things work out OK."*
> >
> > - Right, that makes sense. However, my question then is the following- "why do we need to discuss the online setting at all if the problem occurs due to the abstraction and even when the samples are iid" as you state here - "We disagree with these remarks, what we show in the counterexample is specifically that adding abstraction is what changes the samples to be dependent. In the original MDP, assuming ergodicity, the next states s' for each s,a are independent."
> >
> > *"In the original MDP, assuming ergodicity, the next states s' for each s,a are independent."*
> >
> > - This is precisely what I do not understand. Why does the fact that "the abstraction causes dependence in samples" change when we are given independent states vs dependent states?
> >
> > *"Notation"*
> >
> > This line is where I found the rest of the paper hard to follow from: "We slightly overload notation and let s¯ both denote an abstract state as well as the set of ground states that map to the abstract state." My suggestion would be to us two different variables, $x$ and $s$ to denote the ground and abstract states respectively. This is pretty common in abstraction based papers in RL as well.

---

> > > ### Author Response · Authors · 2021-08-18
> > > **Re: Re: Initial author response**
> > >
> > > *"I see the connection you are trying to make now. This looks interesting but I would argue that this is only so if the abstraction is not Markovian in nature. I do not think that is the case for bisimulation though, i.e. it is Markovian."*
> > >
> > > That is true, when the abstraction is "exact"/Markovian (i.e., a perfect "MDP bisimulation" or "model similarity abstraction" ) then the resulting model *is* an MDP, and none of the problems that we raise occur.
> > >
> > > However, here we deal with an "approximate model similarity abstraction (AMSA)" [Abel16ICML]. So in this case, it is not guaranteed to be exact. And here it is that problems occur: even though there are bounds for such AMSAs, and there are bounds for model-based RL, putting these two together is not trivial, leading to mistakes in the literature.
> > >
> > > *"Right, that makes sense. However, my question then is the following- "why do we need to discuss the online setting at all if the problem occurs due to the abstraction and even when the samples are iid" as you state here - "We disagree with these remarks, what we show in the counterexample is specifically that adding abstraction is what changes the samples to be dependent. In the original MDP, assuming ergodicity, the next states s' for each s,a are independent.""*
> > >
> > > Apologies for failing to have clarified this. Let us try and get this straight. We need to discuss the online setting because it is not always possible to find an offline solution, we may not have a model or simulator of the environment available. And in the online case, we can not sample the underlying states “from each abstract state” independently.
> > >
> > > * In the original MDP, assuming ergodicity, the next state s’ for a particular <s,a> are conditionally independent, given that we arrived in <s,a>, the s’ is drawn i.i.d. s’ ~ T(.|s,a).
> > > * In "the offline setting" (i.e., we know T in advance) we can construct an Abstract MDP, which *is* an MDP with known abstract transition model \bar{T}. This can then be used for dynamic programming or as a simulator for RL methods.
> > > * In "the online setting" (i.e., we do not know T in advance and we don't have a simulator) we can try and directly learn $\bar{T}$ via model-based RL methods. However, we show that even though s' ~ T( . | s,a) are drawn independently and are in fact i.i.d. (for each <s,a>), it is *not* the case that samples
> > > $\bar{s}'$ ~ $\bar{T}( . | \bar{s},a)$
> > > are independent (for each <$\bar{s}$,a> pair).
> > > * This happens because the probability of the next $\bar{s}’$ for a specific state action pair <$\bar{s} = A, a = 1$> can depend on the full history of abstract states and actions, and therefore also on the previous occurrence of <$\bar{s}= A, a=1$>.
> > > I.e., in the counterexample, the probability of the next $\bar{s}’$ for <$\bar{s}= A, a=1$>  depends on the specific $s \in A$ that we reach,
> > > $$P(\bar{s}’ | \bar{s} = A, a = 1) =
> > > P(\bar{s}’| s = 1, a = 1) P(s = 1| history) + P(\bar{s}’| s = 2, a = 1) P(s = 2| history). $$
> > >
> > > *This is precisely what I do not understand. Why does the fact that "the abstraction causes dependence in samples" change when we are given independent states vs dependent states?*
> > >
> > > We are not certain what the confusion is here, and hope the above response has already clarified.
> > >
> > > A point of confusion might be caused by the simulator setting, which perhaps is less trivial than it can appear. Here we have the ability to sample the *abstract* states independently: $\bar{s}'$ ~ $\bar{T}( . | \bar{s},a)$ are independent (which is guaranteed by selecting a prototype s for each $\bar{s}$).
> > >
> > > The crucial property to be able to apply mainstream model-based RL techniques is to guarantee that *abstract states* are sampled independently from the estimated $\bar{T}$. When using "online data collection" this seems very difficult to guarantee.
> > >
> > > *This line is where I found the rest of the paper hard to follow from: "We slightly overload notation and let s¯ both denote an abstract state as well as the set of ground states that map to the abstract state." My suggestion would be to use two different variables,  and  to denote the ground and abstract states respectively. This is pretty common in abstraction based papers in RL as well.*
> > >
> > > We do think our system is reasonably clean: any “s” is a ground state and everything with a \bar is "abstracted”.
> > > Maybe overloading an abstract state $\bar{s}$ to denote both the index of that abstract state, as well as the set of ground states it represents leads to some unnecessary difficulties.
> > > Perhaps, if the reviewers agree, we could introduce more elaborate notation such as $s \in \mathcal{\bar{s}}$ to denote the latter?

---

> > > > ### Comment · Reviewer_qhw2 · 2021-08-19
> > > > **Re: Re: Initial author response**
> > > >
> > > > Ok, I think I understand the contributions much better now. From what I get, you want to say the following:
> > > >
> > > > - When combining state abstraction and MBRL in the case where we do not know the true transition distribution, abstract samples are no longer iid. This has implications when we try to apply any model based method that does an MLE based estimate over states outputed by function approximation (NNs) over observations.
> > > >
> > > > I think the biggest source of confusion has been calling the online setting as the one where we do not have access to the true transition distribution. I do not think that is accurate to say. We can still assume, or not assume, access to the true transition distribution in both the offline and the online collection process. Maybe what you want to say is that if the data is already collected, we can *estimate* the true model first and then build state abstractions over it, so it is close to assuming that we have the true transition distribution. If that is true, I'd argue that the counterexample you show would still apply (i.e. in the offline case) there unless you have a lot of data samples. Let me know what you think of this comment and the summary above and if there is something I am still getting wrong here/misunderstanding something.
> > > >
> > > > Here are other detailed comments, some regarding how the understanding can be improved, and others are some doubts I have:
> > > >
> > > > *That is true, when the abstraction is "exact"/Markovian (i.e., a perfect "MDP bisimulation" or "model similarity abstraction" ) then the resulting model is an MDP, and none of the problems that we raise occur.* **vs** *"However, when we assume that φ is an approximate model similarity abstraction [1] this worst case may not apply"*:
> > > >
> > > > - You might want to edit this line to convey the above sentiment then. I get what you're trying to say here but to a casual reader these two might seem contradictory.
> > > >
> > > > *Abstracted RL: in this setting there is a nonstationarity caused by the clustering of states with different dynamics. There is a lot of related work in other abstraction settings (e.g., state aggregation) where this complication does not occur due to the particularities of their setting*
> > > >
> > > > - This sentence needs editing as well since the setting you consider is exactly of state aggregation based abstraction.
> > > >
> > > > *More comments*
> > > >
> > > > - Line 58: $Y_{s,a}$ is duplicated
> > > > - Line 112: should use $\bar{s}$ instead of s'? In the exp where weights sum to one...
> > > >
> > > > - Eq. 15 uses $Y_{s,a}$ which is supposed to denote the set of next states. Shouldn't you have $\bar{s}'$ in the definition then, instead of $\bar{s}$?. Line 197 should also use $\bar{s}'$ otherwise the example is very difficult to understand.
> > > >
> > > > - In the counter example, you have three terms, 0, 0.6*0.6, and 0.4*0.4. I presume the first one correponds to the first transition being to abstract state C. What do the other two correpond to? The notation you use above doesn't correpond to the one in paper. In the paper, you have $P(\bar{s}' | \bar{s}) P(\bar{s})$ whereas here you use $P(\bar{s}' | s) P(s)$.
> > > >
> > > > - Does the example satisy a bisimulation abstraction? I presume you can define the reward such that the bisimulation condition for reward holds easily but what about the transition condition? $P(\bar{s}' = B | s = 1, a)$ should be equal to $P(\bar{s}' = B | s = 2, a)$ according to bisimulation's transition condition, whereas in the counter example, these are 0.6 and 0.4 respectively.

---

> > > > > ### Author Response · Authors · 2021-08-20
> > > > > **Re: Re: Initial author response**
> > > > >
> > > > > "*Let me know what you think of this comment and the summary above and if there is something I am still getting wrong here/misunderstanding something.*"
> > > > >
> > > > > I guess that there has been some confusion w.r.t. terminology, that we are happy to clarify:
> > > > >
> > > > > --"access to transition function" for us means "knowing the probabilities T(s'|s,a)". This is the critical distinction since if we know the probabilities, we can construct $\bar{T}$ using equation 8. We refer to the setting in which we do not “know the probabilities” as the "the online setting", because we will need to learn about the transition function when acting. If clearer, we would be happy to refer to this as "the setting without knowing the model" or similar.
> > > > >
> > > > > --The setting you describe "Maybe what you want to say is that if the data is already collected, we can *estimate* the true model first and then build state abstractions over it, so it is close to assuming that we have the true transition distribution." is not completely specified:
> > > > > 1. if the collected data are states s of the original MDP, one can first estimate T (and its uncertainty) and then use equation 8 to compute an abstract MDP with transitions $\bar{T}$ (where we could propagate the uncertainty). Since this *is* an MDP, it can now be used as a simulator with existing MBRL methods without any issues (other than additionally taking into account the uncertainty). However, storing unabstracted states s might not be feasible (might require too much data). [Note that we currently call this the "offline setting" essentially the estimated model T is the starting point from our perspective]
> > > > >
> > > > > 2. if collected data are abstracted states $\bar{s}$, then our counterexample shows that a model $\bar{T}$ that would be estimated using existing MBRL methods, might not satisfy the bounds stated for those methods (because the samples $\bar{s}$ are not independent). [We currently call this the online setting: samples $\bar{s}$ have been collected from the true environment, and now we need to act based on this information]
> > > > >
> > > > > "*Here are other detailed comments, some regarding how the understanding can be improved, and others are some doubts I have:*"
> > > > >
> > > > > Thank you for the comments, we will consider making changes to the wording in order to avoid possible confusion.
> > > > >
> > > > > "*- Line 112: should use $\bar{s}$ instead of s'? In the exp where weights sum to one…*"
> > > > >
> > > > > We can see where the confusion comes from, we could rewrite it to something along the lines of “and the weights of the ground states associated with an abstract state $\bar{s} sum to 1: \sum_{s \in \bar{s}} \omega(s,a) = 1$.”
> > > > >
> > > > > "*- Eq. 15 uses $Y_{s,a}$ which is supposed to denote the set of next states. Shouldn't you have $\bar{s}'$ in the definition then, instead of $\bar{s}$?. Line 197 should also use $\bar{s}'$ otherwise the example is very difficult to understand.*"
> > > > >
> > > > > Yes, they are the next states. This is definitely something we will change if this helps the understanding.
> > > > >
> > > > > "*- In the counter example, you have three terms, 0, 0.6*0.6, and 0.4*0.4. I presume the first one correponds to the first transition being to abstract state C. What do the other two correpond to? The notation you use above doesn't correspond to the one in paper. In the paper, you have $P(\bar{s}' | \bar{s}) P(\bar{s})$ whereas here you use $P(\bar{s}' | s) P(s)$.*"
> > > > >
> > > > > They correspond to A, B and C, respectively. Perhaps it would be clearer if we had written this out. Since, starting in abstract state A, we can not directly reach abstract state A and we have that $\Pr(\bar{Y}_A^{(1)} = A) = 0$. ($\bar{Y}_A^{(1)}$ is the first "next state" that we reach from A, it can only be B or C).
> > > > >
> > > > > For the example in the paper $P(\bar{s}' | s) P(s)$ and $P(\bar{s}' | \bar{s}) P(\bar{s})$ work out to the same result. If $P(\bar{s}' | s) P(s)$ would be clearer, we could consider changing it in the paper. And, space allowing, write out the equation in more detail.
> > > > >
> > > > > "*- Does the example satisfy a bisimulation abstraction? I presume you can define the reward such that the bisimulation condition for reward holds easily but what about the transition condition? $P(\bar{s}' = B | s = 1, a)$ should be equal to $P(\bar{s}' = B | s = 2, a)$ according to bisimulation's transition condition, whereas in the counter example, these are 0.6 and 0.4 respectively.*"
> > > > >
> > > > > No, the example satisfies the approximate model similarity abstraction, definition 1, with $\eta \geq 0.2$ (ignoring the reward).
> > > > > Note that if it did satisfy the bisimulation criteria and thus $P(\bar{s}' = B | s = 1, a)$ was equal to $P(\bar{s}' = B | s = 2, a)$, we wouldn’t have a problem. We’d have that $\Pr(\bar{Y}_A^{1} = B) \Pr(\bar{Y}_A^{2} = B) = \Pr(\bar{Y}_A^{1} = B, \bar{Y}_A^{2} = B)$.

---

> > > > > > ### Comment · Reviewer_qhw2 · 2021-08-22
> > > > > > **Great**
> > > > > >
> > > > > > This is starting to make a lot of sense now. Most of my doubts are now cleared and I am almost ready to champion this paper. A few questions still remain based on your last reply:
> > > > > >
> > > > > > *"They correspond to A, B and C, respectively. Perhaps it would be clearer if we had written this out. Since, starting in abstract state A, we can not directly reach abstract state A"*
> > > > > >
> > > > > > Shouldn't these be the next to next state then? $Y^{(1)}_{A}$ is the first transition from state 1, which could only go to B or C.
> > > > > >
> > > > > > Therefore for the next transition, $Y^{(2)}_{A}$ can only land back to state A (from B and C).
> > > > > >
> > > > > > Following this, the next to next, or third transition $Y^{(3)}_{A}$ can again go to B or C. So it should be the next to next transition right? Then the $0.6 \times 0.6 + 0.4 \times 0.4$ make sense.
> > > > > >
> > > > > > *"For the example in the paper and work out to the same result. If would be clearer, we could consider changing it in the paper. And, space allowing, write out the equation in more detail."*
> > > > > >
> > > > > > - I think this should be changed definitely and the equations should be expanded at least by one more intermediate step.
> > > > > >
> > > > > > *"Note that if it did satisfy the bisimulation criteria, we wouldn’t have a problem."*
> > > > > >
> > > > > > - Right this makes sense. I have two follow-up questions here. Approximate model similarity is the same as approximate bisimulation (model-irrelevance), right? Maybe stating this in the paper is a good idea since bisimulation is well situated in the abstractions literature. Furthermore, you say that if it was not exact, we wouldn't have a problem. Therefore, if the 0.4 probability is replaced by 0.6, we should get 0.72 instead of 0.52 in Eq. 17, ending up with 0.432 instead of 0.321. Whereas the joint probablity would still be 0.36. Why would there be such a discrepancy? Anything I'm missing here?
> > > > > >
> > > > > > Lastly, thanks for being patient with the replies so far!

---

> > > > > > > ### Author Response · Authors · 2021-08-23
> > > > > > > **Re: Great**
> > > > > > >
> > > > > > > *"This is starting to make a lot of sense now. Most of my doubts are now cleared and I am almost ready to champion this paper.”*
> > > > > > >
> > > > > > > That is very encouraging. Please let us know if further clarification could help!
> > > > > > >
> > > > > > > *“Shouldn't these be the next to next state then? $Y^{(1)}_{A}$ is the first transition from state 1, which could only go to B or C.*
> > > > > > >
> > > > > > > *Therefore for the next transition, $Y^{(2)}_{A}$ can only land back to state A (from B and C).*
> > > > > > >
> > > > > > > *Following this, the next to next, or third transition $Y^{(3)}_{A}$ can again go to B or C. So it should be the next to next transition right? Then the $0.6 \times 0.6 + 0.4 \times 0.4$ make sense."*
> > > > > > >
> > > > > > > We used the index “A” to imply that they are the abstract next state samples for abstract state A, so $\bar{Y}^{(2)}_{A}$ is the second abstract next state that was reached from abstract state A (after we landed back there from either B or C). We will state this explicitly.
> > > > > > >
> > > > > > > *“- Right this makes sense. I have two follow-up questions here. Approximate model similarity is the same as approximate bisimulation (model-irrelevance), right? Maybe stating this in the paper is a good idea since bisimulation is well situated in the abstractions literature.”*
> > > > > > >
> > > > > > > Yes, you are right, we will include an explanation similar to the following:
> > > > > > > Over the years people have considered equivalence notions in MDP using different terminology. Essentially
> > > > > > > -an (exact) stochastic bisimulation [Givan et al'03AIJ, theorem 7] corresponds to a model similarity abstraction [Li et al. 2006].
> > > > > > >
> > > > > > > -Givan et al. '03 (section 6.2) also describe what we could call an "approximate stochastic bisimulation" by stating "relaxation can be done by allowing the aggregation of states
> > > > > > > into the same “equivalence” class even though their transition probabilities to other
> > > > > > > blocks are different, so long as they are approximately the same (i.e., within $\epsilon$ of each other, for some parameter $\epsilon$ ).". This directly corresponds to the notion of "approximate model similarity" (Def. 13 of Abel et al. 2016 ICML].
> > > > > > >
> > > > > > > *“Furthermore, you say that if it was not exact, we wouldn't have a problem. Therefore, if the 0.4 probability is replaced by 0.6, we should get 0.72 instead of 0.52 in Eq. 17, ending up with 0.432 instead of 0.321. Whereas the joint probablity would still be 0.36. Why would there be such a discrepancy? Anything I'm missing here?”*
> > > > > > >
> > > > > > > In the problem as stated, the shown abstraction is not exact. It would be an exact abstraction for a different MDP where the transition probabilities from state 1 and 2 to state 3 and 4 would be the same.
> > > > > > >
> > > > > > > So let’s say that we set: $\Pr(3|1) = \Pr(3|2) = 0.6$ and $\Pr(4|1) = \Pr(4|2) = 0.4$. Then we’d only replace one of the 0.4 probabilities in equation 17 with 0.6 (since the 0.4*0.4 was the probability $\Pr(B|2) * \Pr(2|C)  * \Pr(C|1) = 0.4 * 1 * 0.4$).
> > > > > > >
> > > > > > >
> > > > > > > Then we get $$\Pr(B|1) * (\Pr(B|A) * \Pr(A|1) + \Pr(B|1) * \Pr(1|B)  * \Pr(B|1) + \Pr(B|2) * \Pr(2|C) * \Pr(C|1)) $$
> > > > > > > $$= 0.6 * (0 + 0.6 * 1* 0.6 + 0.6  * 1 * 0.4) = 0.6 * (0.36 + 0.24) = 0.36.$$
> > > > > > > And the joint probability is indeed also still 0.36.
> > > > > > >
> > > > > > > We hope this clarifies the confusion?
> > > > > > >
> > > > > > > *“Lastly, thanks for being patient with the replies so far!”*
> > > > > > >
> > > > > > > Thank you for the suggestions and for engaging in the discussion!

---

> > > > > > > > ### Comment · Reviewer_qhw2 · 2021-08-23
> > > > > > > > **Thanks**
> > > > > > > >
> > > > > > > > I see. I should have thought more about the exact version of the counter-example. Your explanation makes sense. Overall, I think a lot of the confusion was caused due to the 'online' vs 'offline' settings used in the paper and their relation to knowing the transition function. I think the revised version of the paper should clearly state when the dependence problem occurs, i.e. when we do not assume knowing the transition function + when we collect the data online. This should be followed by a comment emphasizing the importance of not observing the original/ground states (similar to the discussion we had in the comment "The setting you describe is not completely specified:"). The other suggestions that we discussed uptil now will be useful as well in improving the overall understanding. Finally, I think there should be a mention of this in the larger context of neural network like abstractions as well, where these abstractions are followed by learning transition models of the world.
> > > > > > > >
> > > > > > > > I am now updating my score to 7 and do believe this paper should be accepted (with the above noted revisions).

---

### Official Review · Reviewer_drVs · 2021-07-16

**Rating:** 3
**Confidence:** 4

**Summary:**

In this work the authors present a theoretical analysis of the problems which arise when applying model based RL methods to abstracted MDPs. An abstracted MDP is obtained by mapping each state of the original MDP to a new abstract state. Subsequently, a model of the
transition dynamics of this abstracted MDP can be learned. This learned model can be used to solve the abstracted MDP and the resulting policy can be applied to the original MDP. The authors,  examine whether there are any theoretical guarantees regarding the quality of the learned model of the abstracted MDP and of the produced policy. They conclude that collecting data in an online fashion can break the independence assumption made by most MBRL methods in the abstracted case.

**Limitations And Societal Impact:**

Yes

**Main Review:**

The paper examines the interesting problem of learning models of abstracted MDPs. Many popular MBRL methods use deep neural network function approximators to learn models of the MDP dynamics, and can be considered as models of abstracted MDPs.

The analysis presented in the paper separates the abstraction and model learning parts, which is not the case in most real world methods. Moreover, it is not clear how the conclusions drawn by the paper can be used to inform the design of better MBRL algorithms in abstracted MDPs.  The authors show that having access to a perfect simulator to sample independent transitions is necessary in order to have theoretical guarantees, but do not propose any methods for achieving the same effect in the online setting which is mostly used in real-world problems.

There is room for improvement in terms of how the ideas are presented in the paper.

Minor comments:
line 57, 58: Could be rephrased
line 78 - 82: Could be rephrased
Whole paper: Fix the notation of the for every symbol so that the arguments are not subscripts.

**Time Spent Reviewing:**

3

---

> ### Author Response · Authors · 2021-08-09
> **Initial author response**
>
> Thank you for the feedback.
>
> “The paper examines the interesting problem of learning models of abstracted MDPs. Many popular MBRL methods use deep neural network function approximators to learn models of the MDP dynamics, and can be considered as models of abstracted MDPs.
> 	The analysis presented in the paper separates the abstraction and model learning parts, which is not the case in most real world methods.”
>
> In this paper we focus on tabular methods of model learning rather than methods that use deep neural networks. In the tabular case this separation of model learning and abstraction is done fairly often, at least in the literature (e.g. most of the works in our related work section). Our contribution, first and foremost, tries to address an existing problem in this area of the literature. Given that our counterexample occurs even in the tabular case, it also highlights the problem in more general approaches where abstractions (state representations) are learned. We also point out that most real-life applications of MDPs and RL (e.g., in logistics, and other operation research domains) actually are based on human-designed state spaces.
>
> “Moreover, it is not clear how the conclusions drawn by the paper can be used to inform the design of better MBRL algorithms in abstracted MDPs. The authors show that having access to a perfect simulator to sample independent transitions is necessary in order to have theoretical guarantees, but do not propose any methods for achieving the same effect in the online setting which is mostly used in real-world problems.”
>
> Indeed we do not have a good solution yet. It is precisely our paper that makes clear the urgency to provide good approaches for the online setting: Our main contribution however is pointing out a problem in the combination of MBRL and abstraction, that has gone largely unaddressed in the literature.

---

### Official Review · Reviewer_a7tq · 2021-07-16

**Rating:** 5
**Confidence:** 4

**Summary:**

The paper identifies a technical issue when integrating state abstraction and model-based reinforcement learning, namely that obtaining iid samples for the next-state distribution may not be feasible in general. To resolve this issue, the paper suggest, unsurprisingly, that a simulator could be employed.

**Limitations And Societal Impact:**

Yes

**Main Review:**

The paper explores the space in the intersection of model-based RL and abstraction, two key ideas for sample efficiency in RL. A model-based algorithm is defined as one that constructs a transition model of the MDP (could be base or abstracted MDP), and state abstraction is defined, as far as I can say, as an aggregation function.

In order to apply existing model-based RL machinery, it is imperative to get iid samples of the next-state distribution. The nice insight of the paper is to show, concretely, and via a counter example, that such a luxury may not always be available. Intuitively, the fact that the first transition into an abstract state has occurred may give us some clue about the probability of the next transition into the said abstract state, which highlights a dependence between the two samples.

Even though I really value the observation made here, I have two concerns which I list below:

1- The negative result is very narrow. First, it is debatable if the concept of abstraction should just be restricted to state aggregation. It seems to me that the counter example could have been presented for a broader definition of "abstraction", for example, at least get extended to those that probabilistically map to abstract states. Now, even when restricting to the aggregation class, the presented result seems to have a narrow scope. For example, the kind of state abstraction chosen here is obviously not the one and only abstraction function used for RL, or model-based RL for that matter. This particular state abstraction function is, for example, not transitive. Having a low L1 loss is also unnecessarily specific, and could really be generalized to arbitrary f-divergences and/or IPMs, which facilitate the generalization of this result to continuous state spaces (not just tabular problems).

2- The provided remedy, i.e., using a simulator is quite anticlimactic. Notice that the sample efficiency analysis of model-based RL with abstraction may fail for reasons other than iid issue, but if the iid issue is the only hurdle, i believe there are ways to work around the independent samples and still get concentration inequalities using martingales. So I feel like the first half of the paper, namely identifying the problem is nicely done and complete, but a more advanced analysis using independent samples could have really nicely complement the first half.



**Time Spent Reviewing:**

4

---

> ### Author Response · Authors · 2021-08-09
> **Initial author response**
>
> Thank you for the analysis of our paper and valuable feedback.
>
> “1- The negative result is very narrow. First, it is debatable if the concept of abstraction should just be restricted to state aggregation.”
>
> The primary reason for our current focus is that we have encountered this actual problem in this narrow part of the literature. Another reason to choose this particular one was to show that it can go wrong even when the transition functions of the aggregated states are very similar, we will further clarify this in our paper.
>
> We could argue the class is very broad, even functional abstraction (neural networks) can implicitly be seen as a mapping to equivalence classes and thus state aggregation.
>
> “It seems to me that the counter example could have been presented for a broader definition of "abstraction", for example, at least get extended to those that probabilistically map to abstract states. Now, even when restricting to the aggregation class, the presented result seems to have a narrow scope.”
>
> If you increase the class of possible abstractions, our counterexample still stays a counterexample.
> Though it is definitely of interest to study this issue in a broader context, our contribution, first and foremost, tries to address an existing problem. We will however add a discussion on the broader context in which this problem could be considered, based on your suggestion.
>
> “2- The provided remedy, i.e., using a simulator is quite anticlimactic. ... So I feel like the first half of the paper, namely identifying the problem is nicely done and complete, but a more advanced analysis using independent samples could have really nicely complemented the first half.”
>
> We can sympathize and we would have loved, of course, to provide more ingenious approaches to this problem.  However, at this time, we think it is primarily important that the oversight in the current literature is exposed, as this is an immediate problem. In a final version, we can elaborate more on future work.

---

### Decision · Program_Chairs · 2021-09-27

**Decision:**

Reject

**Comment:**

The paper discusses the issue of independence of samples when abstractions are used with online exploration in RL, and the central result is a negative result, that these samples cannot be treated simply as iid and concentration bounds designed for such settings such as Hoeffding's (with union bound to handle L1 error) do not apply.

The authors are noted, however, that formal negative results take the form of impossibility results and/or lower bounds which *do not depend on a specific approach to analysis*. Even better, they should be algorithm independent, or at least apply to a wide range of algorithms. In fact, such negative results in RL are frequently published in top conferences in recent years, and the authors can consult the works of authors like Andrea Zanette, Ruosong Wang, Gellert Weisz, etc., as examples. In contrast, the "negative result" of the current paper is specific to an algorithm (model-based, which is fine as it is simple and representative) and a specific type of approaches to analysis (assuming iid and applying concentration bounds when the conditions are violated). This makes the negative result relatively weak and informal. (Sometimes these informal negative results can be useful and insightful, but they usually appear as a side discussion and cannot serve as the main result of a paper, particularly because the insights they offer are often speculative due to the informal nature of the results.)

Besides, there is a plethora of positive results and analysis techniques that potentially apply to the setting studied in the paper, which the paper fails to discuss. In particular, martingale concentration bounds, law of iterated logs, and a few other techniques come to mind when there is dependence between samples (e.g., when the number of samples is a stopping time that depends on the outcomes of random state transitions); https://arxiv.org/pdf/1708.07367.pdf has used quite a few of such techniques and can be a good reference to study. While it is true that few of existing works consider abstractions, the complication introduced by the abstraction is that different samples from the same abstract state are not identically distributed (the second "i" in "iid"), but advanced concentration bounds (e.g., Freedman's) usually don't assume identically distributed variables and can readily handle them. (Authors also made a similar point in Lemma 3, which directly follows from Hoeffding + union bound.)

Furthermore, one can also think of abstractions as a special case of function approximation: for example, on a given dataset, model-based RL with an abstraction is exactly equivalent to Fitted-Q using a corresponding "piecewise constant" function space. To this end, existing analyses that handle generic function approximation need to be discussed. Instead of proving concentration for each abstract state, you would use uniform convergence argument over the entire function space to handle concentration; a good reference is Antos et al (https://link.springer.com/content/pdf/10.1007/s10994-007-5038-2.pdf) which assumes beta-mixing data. It is true that this paper assumes a fixed behavior policy and does not consider changing policies (the authors emphasized this in post-rebuttal discussions), but there are other more recent results that consider specifically online exploration with generic function approximation; see https://arxiv.org/pdf/1610.09512.pdf and follow-up works.

Finally, a reviewer mentioned the lack of experiments. While that is certainly not a weakness of the paper, the authors can consider this as a way to strengthen their negative result. As of now, it is unclear what happens if one uses the "wrong" algorithm in practice, even if we know it is not theoretically sound. It may perform very well in practice and can be a hard-to-beat baseline compared to more sophisticated techniques that are theoretically sound. If degenerate empirical performance can be demonstrated, the issue will draw more attention and convince the community to treat the problem more seriously.